# Safety and Efficacy of Convalescent Plasma in COVID-19: An Overview of Systematic Reviews

**DOI:** 10.3390/diagnostics11091663

**Published:** 2021-09-11

**Authors:** Massimo Franchini, Fabiana Corsini, Daniele Focosi, Mario Cruciani

**Affiliations:** 1Department of Hematology and Transfusion Medicine, Carlo Poma Hospital, 46100 Mantua, Italy; crucianimario@virgilio.it; 2Santorso Hospital, AULSS7 Pedemontana, 36061 Vicenza, Italy; fabiana.corsini@aulss7.veneto.it; 3North-Western Tuscany Blood Bank, Pisa University Hospital, 56126 Pisa, Italy; daniele.focosi@gmail.com

**Keywords:** COVID-19, convalescent plasma, overview, systematic review, therapy

## Abstract

Convalescent plasma (CP) from patients recovered from COVID-19 is one of the most studied anti-viral therapies against SARS-COV-2 infection. The aim of this study is to summarize the evidence from the available systematic reviews on the efficacy and safety of CP in COVID-19 through an overview of the published systematic reviews (SRs). A systematic literature search was conducted up to August 2021 in Embase, PubMed, Web of Science, Cochrane and Medrxiv databases to identify systematic reviews focusing on CP use in COVID-19. Two review authors independently evaluated reviews for inclusion, extracted data and assessed quality of evidence using AMSTAR (A Measurement Tool to Assess Reviews) and GRADE tools. The following outcomes were analyzed: mortality, viral clearance, clinical improvement, length of hospital stay, adverse reactions. In addition, where possible, subgroup analyses were performed according to study design (e.g., RCTs vs. non-RCTs), CP neutralizing antibody titer and timing of administration, and disease severity. The methodological quality of included studies was assessed using the checklist for systematic reviews AMSTAR-2 and the GRADE assessment. Overall, 29 SRs met the inclusion criteria based on 53 unique primary studies (17 RCT and 36 non-RCT). Limitations to the methodological quality of reviews most commonly related to absence of a protocol (11/29) and funding sources of primary studies (27/29). Of the 89 analyses on which GRADE judgements were made, effect estimates were judged to be of high/moderate certainty in four analyses, moderate in 38, low in 38, very low in nine. Despite the variability in the certainty of the evidence, mostly related to the risk of bias and inconsistency, the results of this umbrella review highlight a mortality reduction in CP over standard therapy when administered early and at high titer, without increased adverse reactions.

## 1. Introduction

The Coronavirus Disease 2019 (COVID-19) pandemic, caused by the Severe Acute Respiratory Syndrome Coronavirus 2 (SARS-CoV-2), is still a worldwide health crisis with devastating social and economic consequences. Despite the virus known for more than a year and a half, the management of COVID-19 remains challenging with still high mortality rates among severely affected patients [1,2]. A number of guidelines from experts have been released during the pandemic period suggesting several treatments for COVID-19 patients, including antiviral, hydroxychloroquine, steroid, anticoagulation and other supportive therapies [3,4]. However, recent evidence from large-scale studies failed to clarify the efficacy of most of the treatments proposed [5,6,7].

Convalescent plasma (CP), first introduced in 1890 by Emil von Behring to treat diphtheria and pertussis and then utilized in several other serious infectious diseases, including Ebola, severe acute respiratory syndrome (SARS), and Middle East respiratory syndrome (MERS), has been also proposed more recently as passive immunotherapy for treatment of SARS-CoV-2 infection [1]. Convalescent plasma is nowadays among the most studied and utilized antibody-based therapies against COVID-19 world-wide and a number of published or ongoing clinical trials have been conducted to assess its efficacy and safety in this challenging viral infection. Their conclusions are, however, quite conflicting and reflect the wide heterogeneity between different studies in terms of CP product, patients enrolled, and disease characteristics. Because of the huge amount of clinical data available, several systematic reviews (SRs) and meta-analysis have been published in the last year to harmonize the results from primary clinical trials. To synthesize the evidence from these SRs and meta-analyses, we have decided to apply to this clinical setting a relatively new approach, i.e., to perform an overview of the existing SRs, also called umbrella review.

## 2. Material and Methods

This umbrella review was registered at the International Prospective Register of Systematic Reviews (PROSPERO) with the registration number CRD42021259625.

### 2.1. Review Question/Objective

The aim of this umbrella review is to evaluate the efficacy and safety of CP for the treatment of COVID-19 patients.

### 2.2. Inclusion and Exclusion Criteria

We considered for inclusion SRs that included randomized controlled trials (RCTs) and non-RCTs (i.e., prospective, retrospective, cross-sectional, cohort studies and case series) evaluating the safety and efficacy of CP in COVID-19 patients. Reviews without qualitative and/or quantitative analysis were excluded from this umbrella review. SRs evaluating other viral infections were excluded unless they also contained data on SARS-COV-2 infection that could be evaluated separately.

### 2.3. Clinical Setting and Participants

For this umbrella review, we considered SRs on COVID-19 at any stage of disease severity, from asymptomatic/paucisymptomatic to life-threatening cases, and in any setting (outpatients and hospitalized patients). In addition, we included populations of patients with no limitations of age, gender, ethnicity, or comorbidities.

### 2.4. Intervention and Outcomes

CP treatment at any titer, dose, timing and frequency was compared to standard of care or placebo. We included the following outcomes: all-cause mortality, viral clearance, clinical improvement, length of hospital stay, serious and non-serious adverse reactions. Subgroup analyses were also performed based on the severity (i.e., non-severe versus severe) of COVID-19 patients treated with CP and on the titer (i.e., high versus low titer) and timing (i.e., <3 days versus >3 days of hospital admission) of CP transfusion.

### 2.5. Search Strategy

Relevant studies in four bibliographic databases (Embase, PubMed, Web of Science, and Cochrane) and a preprint database MedRix were searched as of 15 August 2021. The searches were carried-out without languages restriction using Medical Subjects Heading: (“COVID-19” OR “SARS-CoV-2”) AND (“convalescent plasma” OR “serotherapy” OR “hyperimmune plasma therapy” OR “convalescent plasma treatment”) AND (“systematic review” OR “meta-analysis”). In addition to the electronic search, we checked the reference lists of the most relevant items (original studies and reviews) to identify potentially eligible studies not captured by the initial literature search.

### 2.6. Study Selection and Data Extraction

All titles were screened by two independent assessors (MC and MF). Eligibility assessment was based on the title or abstract and on the full text if required. Full texts of possibly eligible articles were obtained and assessed independently by two reviewers (MC and MF). Both reviewers compared the articles identified. Studies were selected independently by two reviewers (MF and MC) with disagreements resolved through discussion and on the basis of the opinion of a third reviewer (FC). The two assessors also independently extracted quantitative and qualitative data from each selected study. Findings are presented in tabular format with supporting text (Table 1). Quantitative tabulation of results include: first author name and year of publication, the clinical condition under evaluation, principal characteristics of the study population, number of RCTs and non-RCTs included in the SR, intervention (CP versus control characteristics), the outcomes assessed, a quantitative synthesis (when available) of the estimates of interest (odds ratio (OR), risk ratio (RR), risk difference (RD), or mean difference (MD) with the 95% confidence intervals (CI), as reported in individual reviews), and the main results and conclusions of the SR. In addition, a three-color score was used for an immediate visual inspection of the CP-related effects with regard to the four main outcomes assessed: viral clearance, clinical improvement, mortality reduction and safety (green color: CP confers advantage over standard therapy or placebo; red color: CP does not confer advantage over standard therapy or placebo; yellow color: no clear advantage or disadvantage).

### 2.7. Assessment of Methodological Quality

We used A Measurement Tool to Assess Reviews (AMSTAR-2) critical appraisal checklist for SRs, a tool that evaluates both quantitative and qualitative reviews [8]. The tool is suitable for reviews including randomised and non-randomised studies. It includes 16 domains relating to the research question, review design, search strategy, study selection, data extraction, justification for excluded studies, description of included studies, risk of bias, sources of funding, meta-analysis, heterogeneity, publication bias, and conflicts of interest. Two review authors (MC, MF) independently assessed the quality of evidence in the included reviews and the methodological quality of the SRs. We resolved discrepancies through discussion or, if needed, through a third review author (FC). We did not exclude reviews based on AMSTAR-2 ratings, but considered the ratings in interpretation of our results.

### 2.8. Appraisal of the Quality of Evidence

The quality of evidence was appraised following the Grades of Recommendation, Assessment, Development, and Evaluation (GRADE) approach. Whenever available, the grading of the quality of evidence reported in the included reviews was considered to determine the quality of evidence. In a situation in which the grading of evidence was not reported by the authors of the study, the GRADE approach was applied in its five domains (risk of bias, indirectness, imprecision, inconsistency, and publication bias) based on the information available from the study [9].

**Table 1 diagnostics-11-01663-t001:** Summary of the published systematic reviews and meta-analyses.

First Author	Clinical Setting	Population	Studies Included in Quantitative Analysis (n)			Intervention		Outcomes	Main Results
			Overall (Patients)	RCT	NRCT	CP	Control		
Aviani [10]	MERS-CoV, SARS-CoV-1, SARS-CoV-2 and influenza infections (severe or critically ill patients)	Patients infected with beta coronaviruses or influenza viruses with no limitations of age, gender or ethnicity	20	5	15	CP at any titer, dose, timing and frequency	Standard care	30-day mortality, safety, viral clearance, clinical improvement, discharge rate	CP significantly reduced mortality and increased the number of discharged patients. Less than 1% of serious transfusion-related AEs
Bansal [11]	COVID-19 at any stage of disease severity	Adult (>18 years) patients hospitalized for COVID-19	23 (27,706)	10	13	CP at any titer, dose, timing and frequency	Placebo or standard therapy	Mortality, safety	CP significantly reduced mortality. A 6.1% pooled AE rate was observed with no CP-related fatalities
Barreira [12]	Moderate, severe or critical COVID-19	Hospitalized COVID-19 patients with no limitations of age, gender or ethnicity	11 (3098)	5	6	CP at any titer, dose, timing and frequency	Placebo or standard care	Mortality, safety, viral clearance, clinical improvement, length of hospitalization	CP significantly reduced mortality and viral load at 72 h after CP transfusion. A 3.5% AE rate occurred
De Candia [13]	COVID-19 at any stage of disease severity	Patients with laboratory confirmed COVID-19	25 (22,591)	10	15	CP at any titer, dose, timing and frequency	Standard therapy	Mortality	CP, independently from neutralizing titer, significantly reduced mortality only when administered at early stage of the disease
Elbadawi [14]	COVID-19 pneumonia	COVID-19 patients	6 (1226)	6	0	CP at any titer, dose, timing and frequency	Standard care	All-cause mortality, progression to severe respiratory illness, clinical improvement, need for IMV, safety	CP did not reduce all-cause mortality, the progression to sever respiratory illness or the need for IMV. No differences in clinical improvement and AEs
Gupta [15]	COVID-19 at any stage of disease severity	COVID-19 patients	12 (13,206)	12	0	CP at any titer, dose, timing and frequency	Placebo or standard care	28-day mortality, clinical improvement, viral clearance, safety	CP was not associatd with clinical improvement or significantly reduced risk of death. A low incidence (3.2%) of AEs was observed
Janiaud [16]	COVID-19 at any stage of disease severity	Patients with confirmed or suspected COVID-19 in any treatment setting	10 (11,782)	10	0	CP at any titer, dose, timing and frequency	Placebo + standard therapy or standard therapy	Clinical improvement, all-cause mortality, no. of patients requiring IMV, rate of serious adverse events	CP did not reduce the need for IMV and all-cause mortality. A subgroup analysis showed a mortality reduction in patients receiving high-titer CP [17]
Juul [18]	COVID-19 at any stage of disease severity	COVID-19 patients with no limitations of age, gender or ethnicity	33 (13,312)	33 (2 CP)	0	Various interventions for treatment of COVID-19, including CP, were analyzed	Standard care	All-cause mortality, admission to ICU, need for IMV, non-serious AEs	CP did not reduce all-cause mortality
Keikha [19]	MERS-CoV (2 studies), SARS-CoV-1 (5 studies), SARS-CoV-2 (8 studies) infections	Patients infected with beta coronaviruses	15 (5240)	1	14	CP at any titer, dose, timing and frequency	Placebo or standard therapy	Clinical improvement, viral clearance, hospital discharge mortality	The clinical improvement was significantly increased in CP group versus control group
Kim [20]	Moderate-severe COVID-19	Adult (> 18 years) patients hospitalized for COVID-19	110 (4 CP)	40 (2 CP)	70 (2 CP)	Various pharmacological interventions against COVID-19, including CP, were analyzed	Placebo or standard care	Mortality, progression to severe disease, viral clearance, serious adverse events	CP was associated with significantly reduced mortality rate in non-ICU setting and improved viral clearance rate at 2 weeks compared to standard care
Klassen [21]	COVID-19 at any stage of disease severity	COVID-19 patients with no limitations of age, gender or ethnicity	30	10	20	CP at any titer, dose and timing (<3 days versus >3 days of hospital admission)	Standard care	Mortality	CP significantly reduced mortality rate compared with standard care. CP transfusion within 3 days of hospital admission resulted in greater mortality reduction. A subgroup analysis documented the safety of CP [22]
Kloypan [23]	COVID-19 at any stage of disease severity	COVID-19 patients	47	14	33	CP at any titer, dose, timing and frequency	Placebo or standard treatment	28-day mortality, length of hospital stay, clinical improvement, discharge rate	CP significantly reduced the risk of all-cause mortality compared to standard care
Meher [24]	Moderate-severe COVID-19	Hospitalized adult patients of both gender with moderate to severe COVID-19	6 (474)	2	4	CP at any titer, dose, timing and frequency	Standard care	Mortality, clinical improvement, viral clearance	CP significantly reduced all-cause mortality and viral detection by day 7 with no clinical improvement by day 7
Peng [25]	Prophylaxis and treatment of COVID-19	COVID-19 patients with no limitations of age, gender, ethnicity or underlying diseases	13 (2984)	2	11	CP at any titer, dose, timing and frequency	Placebo or standard care	Mortality, clinical improvement, safety	CP significanlty reduced mortality and increased viral clearance
Piechotta [26]	COVID-19 at any stage of disease severity (asymptomatic or symptomatic)	COVID-19 patients with no limitations of age, gender or ethnicity	13 (48,509)	12	1	CP at any titer, dose, timing and frequency	Placebo or standard care	All-cause mortality, clinical improvement, need for IMV, viral clearance, safety	CP did not reduce the need for IMV and 28-day all-cause mortality, but increased 7-day viral clearance. In a subgroup analysis, CP decreased disease progression and all-cause mortality in individuals with asymptomatic or mild COVID-19
Prasad [27]	Severe and non-severe COVID-19	Hospitalized COVID-19 patients with no limitations of age, gender or ethnicity	22 (9622)	9	13	CP at any titer, dose, timing and frequency	Placebo or standard care	Mortality, clinical improvement, need for IMV, viral clearance, length of ICU or hospital stay, safety	Inconclusive effects of CP on mortality, clinical improvement, need for mechanical ventilation and faster viral clearance
Rabelo-da-Ponte [28]	COVID-19 at any stage of disease severity	COVID-19 patients with no limitations of age, gender or ethnicity	9 (149)	1	8	CP at any titer, dose, timing and frequency	Standard treatment	Viral clearance, clinical improvement	CP reduced viral load and was associated with clinical status improvement
Sarkar [29]	COVID-19 at any stage of disease severity	COVID-19 patients with no limitations of age, gender or ethnicity	7 (5444)	2	5	CP at any titer, dose, timing and frequency	Standard treatment	Mortality, viral clearance, clinical improvement	CP reduced mortality, increased viral clearance and was associated with clinical improvement
Sun [30]	Different types of infectious diseases including severe COVID-19	Patients with viral infections with no limitations of age and sex	15	3 (1 CP)	12	CP at any titer, dose, timing and frequency	Standard treatment	Mortality, symptom duration, hospital length of stay, antibody levels, viral load, adverse events	There was a significantly lower mortality rate in the group treated with CP compared with control groups
Talaie [31]	COVID-19 at any stage of disease severity	Adult COVID-19 patients	26 (3263)	14 (1 CP)	12 (5 CP)	Various pharmacological interventions against COVID-19, including CP, were analyzed	Standard treatment	Mortality, viral clearance, clinical improvement, ICU entry, need for IMV	CP had a beneficial effect on clinical improvement and negative seroconversion and tended to decrease mortality
Vegivinti [32]	COVID-19 at any stage of disease severity	COVID-19 patients with no limitations of age, gender or ethnicity	15 (4898)	5	10	CP at any titer, dose, timing and frequency	Standard care	Mortality, clinical improvement, length of hospital stay	CP was associated with a significantly reduced mortality and higher clinical improvement
Wang [33]	COVID-19 at any stage of disease severity	Adult (>18 years) COVID-19 patients	42 (8 CP)	10 (1 CP)	32 (7 CP)	CP at any titer, dose, timing and frequency	Standard care	Mortality, viral clearance	CP tended to decrease the mortality risk and was associated with a higher viral nucleic acid negative rate
Wang [34]	COVID-19 at any stage of disease severity	COVID-19 patients with no limitations of age, gender or ethnicity	45 (44,068)	4	41	CP at any titer, dose, timing and frequency	Placebo, standard care, no intervention	Mortality, clinical improvement, safety	CP reduced (NRCTs) or not (RCTs) mortality and improved (NRCTs) or not (RCTs) clinical symptoms
Wardhani [35]	Mild and severe COVID-19	COVID-19 patients	12 (5342)	3	9	CP at any titer, dose, timing and frequency	Standard care	All-cause mortality, subgroup analysis based on disease severity	All and severe COVID-19 patients not receiving CP were at increased mortality risk compared to those treated with CP
Wenjing [36]	Severe and critical COVID-19	Severely and critically ill COVID-19 patients with no limitations of age, gender or ethnicity	9	3	6	CP at any titer, dose, timing and frequency	NA	Mortality, clinical improvement, safety	CP significantly reduced mortality. A qualitative analysis showed a beneficial effect of CP in reducing viral load, on clinical improvement and on safety
Yuwono Soeroto [37]	COVID-19 at any stage of disease severity	COVID-19 patients	18 (5658)	7	11	CP at any titer, dose, timing and frequency	Standard care	Mortality	CP use was associated with significantly decreased mortality
Zhang [38]	Severe COVID-19	Critically ill COVID-19 patients with no limitations of age, gender or ethnicity	19	1	18	CP at any titer, dose, timing and frequency	Placebo or standard care	Mortality, safety, viral clearance	A significantly reduced mortality rate and a higher negative rate of PCR was found in the CP versus control group

Abbreviations: RCT, randomized controlled trial; NRCT, non-randomized controlled trial; CP, convalescent plasma; NA, not available; IMV, invasive mechanical ventilation; Severe acute respiratory syndrome coronavirus: SARS; Middle East respiratory syndrome: MERS; ICU, intensive care unit; PCR, polymerase chain reaction; AE, adverse event.

## 3. Results

The electronic and manual search retrieved 244 references. At the first stage of screening titles and abstracts, 52 references were selected. The Preferred Reporting Items for Systematic Reviews and Meta-Analyses (PRISMA) flow diagram is reported in Figure 1. After the full texts were scrutinized against the inclusion and exclusion criteria, 29 SRs were included in the umbrella review [10,11,12,13,14,15,16,17,18,19,20,21,22,23,24,25,26,27,28,29,30,31,32,33,34,35,36,37,38] and 23 SRs were excluded [39,40,41,42,43,44,45,46,47,48,49,50,51,52,53,54,55,56,57,58,59,60,61]. Reasons for exclusion were: SRs not covering or with no informative data on CP therapy in COVID-19 [39,40,41,42,44,45,47,48,49,54] and reviews on CP therapy in COVID-19 with no quantitative and/or qualitative analysis [43,46,50,51,52,53,55,56,57,58,59,60,61].

### 3.1. Description of the Studies

Of the 29 SRs included in the overview, 26 were focused exclusively on COVID-19 [11,12,13,14,15,16,17,18,20,21,22,23,24,25,26,27,28,29,31,32,33,34,35,36,37,38], while three were focused on respiratory pandemics and on beta coronaviruses infections [10,19,30]. Two SRs [17,22] were a subgroup analysis of other reviews [16,21]. The 29 SRs included 653 overlapping reports (237 RCTs and 416 non-RCTs), based on 53 individual primary studies. The primary studies included 17 RCTs, 26 controlled non-RCTs, and 10 uncontrolled studies (single arm studies, including case series and case reports). Twenty-eight SRs were focused on CP treatment of COVID-19, while one study [25] was focused on both CP prophylaxis and treatment. The majority of the SRs analyzed COVID-19 at any stage of disease severity, while four studies [10,30,36,38] were focused on more advanced (severe and/or critical) stages. Most SRs analyzed COVID-19 patients with no limitation of age, gender or ethnicity, while four SRs were focused only on adult (>18 years) COVID-19 patients [11,20,31,33]. The main characteristics of the SRs included are summarized in Table 1.

### 3.2. Methodological Quality

Of the included reviews, the majority (19/29; 65.5%) had ≥2 (from 2 to 12) unmet AMSTAR-2 methodological requirements, and nine (31.0%) had one unmet methodological requirement; one Cochrane review met all the methodologic requirements (Table 2). Sixteen reviews (55.2%) had one or more methodological requirements partly met. Twenty-seven reviews (93.1%) did not report on the source of funding for the studies included in the review; 11 reviews (39.2%) did not register or publish a protocol. Eight reviews (27.6%) did not mention publication bias in methods and results, and failed to discuss the possible impact of publication bias on review findings. In five reviews, participants, interventions, comparators, and outcomes (PICO) were not clearly made explicit, and in 3 reviews design was not fully explained. In six reviews, the search strategy was not comprehensive. More than 85% of author teams perform study selection and screening in duplicate. Other unmet domains were related to the list of excluded reviews and reasons (seven reviews, 24.1%), assessment of risk of bias (three reviews), and sources of conflict of interest, including any funding that the authors receive for conducting the review (four reviews).

### 3.3. Summary of the Effect of CP on the Main Outcomes

#### 3.3.1. (a) Outcome “Overall Mortality”

Overall mortality was the most common reported outcome. Great heterogeneity was ascertained in several SRs; thus, we performed subgroup analyses to control for sources of heterogeneity such as design of studies included in the review (e.g., RCTs and non-RCTs), titre of SARS-CoV-2 neutralizing antibodies (high or low), and time of administration (early or late). The results of our analyses are summarized in Table 3. Sixteen SRs reported the outcome mortality in RCTs and non-RCT. In 14 of these SRs the effect size favoured the CP arm compared to controls, while in two SRs it was unclear whether CP reduced mortality compared to controls; the quality of the evidence was very low in 2 SRs, low in four, moderate in nine, and from moderate to high in one. Eleven SRs analysed the outcome mortality in RCTs only, and all were consistent in concluding that CP did not reduce mortality compared to controls (moderate certainty of evidence in 10 SRs, from moderate to high certainty in one SR). The analysis of results from non-RCTs showed higher reduction in mortality in CP group compared to control in six out of seven SRs (low quality of evidence in four, very low in one, moderate in two).

Due to the clinical heterogeneity observed in many of the primary studies and statistical heterogeneity in many of the SRs, we performed subgroup analyses as specified in the protocol. High titre CP was more effective than lower titre in reducing mortality (moderate certainty of evidence in three SRs, low in one). In two SRs, it was unclear whether high titre CP reduces mortality compared to controls (low certainty of evidence). Subgroup analysis according to time to CP transfusion showed a reduction in mortality in early CP recipients compared to controls in three SRs (moderate certainty of evidence); by contrast a Cochrane review based on a single RCT (RECOVERY trial) [62] concluded that it is unclear whether early CP transfusion reduces mortality compared to controls (low quality of evidence), although transfusion after the first week of illness resulted in higher risk of mortality compared to early transfusion. We also performed subgroup analysis of mortality according to baseline severity of COVID-19, but there was heterogeneity in defining the clinical condition. In patients with moderate/severe infection, the effect of CP was unclear (three SRs, low-quality of evidence), while in patients with severe/critical infection, the results were more heterogeneous in the comparison, effect size and certainty of the evidence (Table 3).

#### 3.3.2. (b) Outcome “Viral Clearance”

The outcome viral clearance (rate of patients with negative reverse transcription polymerase chain reaction (RT-PCR) test for SARS-CoV-2 after a positive test at baseline) was reported in 6 SRs after 3 days, and in two SRs after 7–14 days (Table 4). On day 3, four out of six SRs reported an increase in viral clearance in CP recipients compared to controls (from low to moderate certainty of evidence), while in two SRs it was unclear whether CP reduced the viral clearance compared to controls (from low to moderate certainty of evidence). On day 14, CP showed significantly higher viral clearance rate compared to standard supportive therapy in one SR (low certainty of evidence), while in a Cochrane review based on two RCTs it was unclear whether CP increases viral clearance compared to controls (low certainty of evidence). It was not possible to perform subgroup analyses for these outcomes due to the relatively low number of primary studies and SRs available.

#### 3.3.3. (c) Outcome “Clinical Improvement”

Clinical improvement was reported in nine SRs. Six SRs concluded that it is unclear whether CP increases rate of clinical improvement compared to controls (in three SRs moderate certainty of evidence, in three low certainty of evidence). Three SRs showed that CP increase rate of clinical improvement compared to controls (low certainty of evidence in two SRs, moderate certainty of evidence in one SR) (see Table 4).

#### 3.3.4. (d) Outcome “Length of Hospital Stay”

Length of hospital stay was reported in six SRs. All concluded that it is unclear whether CP decreases length of hospital stay compared to controls (low quality of certainty in four SRs, moderate certainty of evidence in two SRs) (Table 4).

#### 3.3.5. (e) Outcome “Adverse Events”

Serious adverse events to CP transfusion were reported in five SRs and overall adverse events in two SRs. All SRs concluded that the frequency of adverse reactions was similar in CP and control groups (low quality of certainty in four SRs, moderate certainty of evidence in one SR) (Table 5).

### 3.4. GRADE Assessment

Of the 89 analyses on which GRADE judgements were made, effect estimates were judged to be of high/moderate certainty in four analyses, moderate in 38, low in 38, and very low in nine. For the outcome mortality, the judgment was very low in nine analyses, low in 19, moderate in 19 and moderate/high in four.

## 4. Discussion

The outbreak of the COVID-19 pandemic has greatly accelerated the clinical trial research evaluating the safety and efficacy of CP as emergency therapy. According to the study by Muller-Olling and colleagues [63], CP is the second most frequent investigational medicinal product evaluated in COVID-19-related clinical trials and increasing interest in this form of immunotherapy is documented by the fact that more than 140 clinical trials specifically evaluating CP in COVID-19 have been registered to date worldwide [63]. Studies on CP as treatment for COVID-19 have qualitatively evolved during the COVID-19 pandemic period in response to the advances in the knowledge of this disease and to the results of the published clinical trials [64]. They can be generally classified into three generations: the first-generation clinical trials, performed at the beginning of the first pandemic wave, utilized CP with high-titer anti-SARS-CoV-2 neutralizing antibodies usually in hospitalized patients with severe, advanced COVID-19 [65,66]. However, it soon became evident that, since CP was working by blocking the viral replication, the earlier it was used the more effective it was in preventing the disease progression [1]. Thus, second generation clinical trials were focused on the use of high-titer CP early in the course of COVID-19 (within 3 days from symptom onset or hospitalization) [67]. Finally, to optimize the possible beneficial effect of plasma, the more recent third generation trials are evaluating the CP infusion in particular populations of patients at high risk of development of severe or critical COVID-19, such as those with impaired humoral immunity, onco-hematological disorders or other severe cardiovascular or respiratory co-morbidities [68,69,70]. Moreover, some of these trials encompass the pre-transfusion evaluation of recipients’ anti-SARS-CoV-2 antibody levels in order to capture those patients with lack or insufficient antibody response and therefore most likely to benefit from CP passive immunotherapy [1]. Based on the above, it is evident that the clinical trials conducted during the 18-month COVID-19 pandemic are widely heterogeneous in terms of study design and CP administration schedule, disease and patients characteristics. In this extremely uncertain and changing context, typical of emergency situations such as those of the COVID-19 pandemic, it is evident that even the systematic reviews and meta-analyses have produced heterogeneous results. This overview of reviews includes data from twenty-nine systematic reviews, based on more than 600 overlapping reports and 53 individual primary studies (43 controlled trials, including 17 RCTs and 26 non-RCTs, and 10 uncontrolled trials (single arm studies). We believe that makes this the largest review to date within this subject area, and hope this will make it particularly helpful to decision makers. The results of RCTs are not always consistent with the results of observational studies, and differences in estimated magnitude of treatment effect are very common, often resulting in overestimation of treatment effects in observational studies [71]. Interpretation of the results obtained from both RCTs and observational studies, as well as from systematic reviews including both types of study design, can help understand the efficacy/effectiveness and safety of a therapeutic option [72]. For this reason we performed, where available, subgroup analyses of the effect size obtained in the overall comparison, in RCTs and in observational studies. For the outcome most commonly reported, overall mortality, it was possible to perform subgroup analysis of SRs according to study design, antibodies titre and time of transfusion. While the majority of SRs reporting this outcome in non-RCTs and in non-RCTs + RCTs showed a reduction in mortality in CP recipients compared to controls, when the analysis was limited to RCTs it was unclear whether CP reduced mortality compared to controls. It was also clear that most of the included studies (both RCTs and non-RCTs) were at risk of bias and showed significant clinical, methodological and statistical heterogeneity. Overall the certainty of the evidence was from low to moderate in the majority of the SRs. Subgroup analyses according to neutralizing antibody titres and time of CP transfusion showed a reduction in mortality in the majority of SRs when high titres of antibody and early transfusions were administered (from low to moderate certainty of the evidence). The other secondary outcomes (i.e., viral clearance, clinical improvement and length of hospital stay) were addressed by only a minority of SRs with a high level of uncertainty, so that no definitive conclusions can be drawn. However, CP seemed to be effective in increasing viral clearance as compared with standard therapy, particularly within the first three days from CP transfusion (from low to moderate certainty of the evidence). Additionally, CP did not increase the risk of adverse events between intervention and control groups, confirming the safety of this procedure [22]. Although the SRs tried to address, at least in part, the heterogeneity of the results (on the basis of neutralizing titre, time of CP infusion and study design) it was almost impossible to evaluate in this overview the clinical heterogeneity of primary studies, related to different disease conditions at baseline, concomitant therapies, patients characteristics. Limitations to the methodological quality of reviews most commonly related to absence of a protocol (11/29) and funding sources of primary studies (27/29).

In conclusion, despite these limitations and based on the analysis of the main outcome mortality, this overview of systematic reviews supports the safety and efficacy of the clinical use of CP over standard therapy when administered at high titer and early during the course of COVID-19. Further pooled qualitative and quantitative analyses from a new systematic review based on individual patients data rather than on aggregate data (that are often insufficient for a thorough analysis) or from adequately powered third generation clinical trials (i.e., assessing the early use of high titer CP administered in populations of patients with inadequate antiviral response and at increased risk of developing severe COVID-19) are needed to pinpoint exactly where and when CP can give the greatest clinical benefit in COVID-19.

## Figures and Tables

**Figure 1 diagnostics-11-01663-f001:**
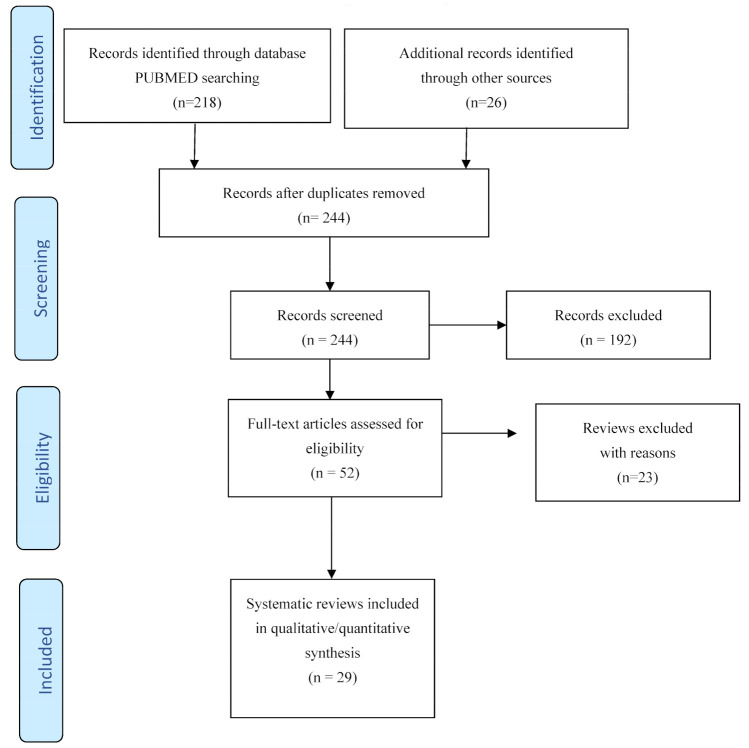
PRISMA flow diagram of study selection.

**Table 2 diagnostics-11-01663-t002:** The AMSTAR-2 checklist.

Author [Reference]	AMSTAR-2 DOMAIN															
	1	2	3	4	5	6	7	8	9	10	11	12	13	14	15	16
Aviani [10]																
Bansal [11]																
Barreira [12]																
De Candia [13]																
Elbadawi [14]																
Gupta [15]																
Janiaud [16]																
Cruciani [17]																
Juul [18]																
Keikha [19]																
Kim [20]																
Klassen [21]																
Franchini [22]																
Kloypan [23]																
Meher [24]																
Peng [25]																
Piechotta [26]																
Prasad [27]																
Rabelo-da-Ponte [28]																
Sarkar [29]																
Sun [30]																
Talaie [31]																
Vegivinti [32]																
Wang M [33]																
Wang Y [34]																
Wardhani [35]																
Wenjing [36]																
Yuwono Soeroto [37]																
Zhang [38]																

Footnotes: 
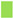
 Methodological requirement met, 
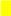
 Methodological requirement partly met, 
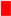
 Methodological requirement unmet. AMSTAR-2 domains: 1. Did the research questions and inclusion criteria for the review include the components of PICO? 2. Did the report of the review contain an explicit statement that the review methods were established prior to the conduct of the review and did the report justify any significant deviations from the protocol? 3. Did the review authors explain their selection of the study designs for inclusion in the review? 4. Did the review authors use a comprehensive literature search strategy? 5. Did the review authors perform study selection in duplicate? 6. Did the review authors perform data extraction in duplicate? 7. Did the review authors provide a list of excluded studies and justify the exclusions? 8. Did the review authors describe the included studies in adequate detail? 9. Did the review authors use a satisfactory technique for assessing the risk of bias (RoB) in individual studies that were included in the review? 10. Did the review authors report on the sources of funding for the studies included in the review? 11. If meta-analysis was performed did the review authors use appropriate methods for statistical combination of results? 12. If meta-analysis was performed, did the review authors assess the potential impact of RoB in individual studies on the results of the meta-analysis or other evidence synthesis? 13. Did the review authors account for RoB in individual studies when interpreting/ discussing the results of the review? 14. Did the review authors provide a satisfactory explanation for, and discussion of, any heterogeneity 15. If they performed quantitative synthesis did the review authors carry out an adequate investigation of publication bias (small study bias) and discuss its likely impact on the results of the review? 16. Did the review authors report any potential sources of conflict of interest, including any funding they received for conducting the review? Although AMSTAR-2 consists of 16 items, critical domains include items 2, 4, 7, 9, 11, 13, and 15.

**Table 3 diagnostics-11-01663-t003:** Effects of convalescent plasma on overall mortality.

Review [Reference]	No. Studies	No. Subjects (CP/Controls)	Effect Size (RR, OR or RD) and 95% CIs	GRADE Assessment	Comment	Effect Direction
Outcome: Mortality	Overall Analysis (RCTs + non-RCTs)					
Aviani [10]	12 (4/8)	4306 (724/3582)	RR 0.62 (0.46/0.82)	⊕⊕⊝⊝ Low 1	CP reduces mortality compared to controls	
Bansal [11]	23 (11/12)	7542 (2392/5150)	OR 0.65 (0.53/0.80)	⊕⊕⊝⊝ Low 1	CP reduces mortality compared to controls	
Barreira [12]	11 (5/6)	2998 (823/2175)	RR 0.71 (0.57/0.90)	⊕⊕⊝⊝ Low 1	CP reduces mortality compared to controls	
De Candia [13]	25 (10/15)	24772 (13470/11302)	RR 0.78 (0.68/0.90)	⊕⊕⊝⊝ Low 1	CP reduces mortality compared to controls	
Klassen [21]	30 (10/20)	12982 (1425/11467)	OR 0.58 (0.47/0.71)	From moderate ⊕⊕⊕⊝ to high ⊕⊕⊕⊕ 3	CP reduces mortality compared to controls	
Kloypan [23]	20 (11/9)	16533 (7753/8780)	RR 0.69 (0.56/0.86)	⊕⊕⊝⊝ Low 1	CP reduces mortality compared to controls	
Meher [24]	6 (2/4)	469 (167/302)	RR 0.61 (0.37/0.99)	⊕⊝⊝⊝ Very low 4	CP reduces mortality compared to controls	
Peng [25]	13 (4/9)	2984 (695/2289)	OR 0.48 (0.34/0.67)	⊕⊕⊝⊝ Low 1	CP reduces mortality compared to controls	
Sarkar [29]	7 (2/5)	5454 (5169/285)	OR 0.44 (0.25/0.77)	⊕⊕⊝⊝ Low 1	CP reduces mortality compared to controls	
Talaie [31]	3 (1/2)	163 (82/81)	RR 0.52 (0.26/1.03)	⊕⊝⊝⊝ Very low 5	It is unclear whether CP reduces mortality compared to controls	
Vegivinti [32]	15 (5/10)	4858 (2208/2650)	OR 0.58 (0.44/0.78)	⊕⊝⊝⊝ Very low 4	CP reduces mortality compared to controls	
Wang Y [34]	3 (1/2)	319 (97/222)	RR 0.65 (0.42/1.02)	⊕⊝⊝⊝ Very low 5	It is unclear whether CP reduces mortality compared to controls	
Wardhani [35]	12 (3/9)	5342 (1937/3405)	OR 1.92 (1.33/2.77)	⊕⊕⊝⊝ Low 1	Mortality higher in controls	
Wenjing [36]	10 (3/7)	2835 (2271/564)	RR 0.57 (0.44/0.74)	⊕⊝⊝⊝ Very low 4	CP reduces mortality compared to controls	
Yuwono Soeroto [37]	18 (6/12)	5657 (2168/3489)	OR 0.64 (0.49/0.84)	⊕⊕⊝⊝ Low 1	CP reduces mortality compared to controls	
Zhang [38]	4 (1/3)	182 (87/95)	RR 0.59 (0.37/0.94)	⊕⊝⊝⊝ Very low 5	CP reduces mortality compared to controls	
	RCTs only					
Bansal [11]	10	1413 (770/653)	OR 0.75 (0.53/1.08)	⊕⊕⊕⊝ Moderate 2	It is unclear whether CP reduces mortality compared to controls	
Barreira [12]	5	1065 (595/470)	OR 0.85 (0.62/1.16)	⊕⊕⊕⊝ Moderate 2	It is unclear whether CP reduces mortality compared to controls	
De Candia [13]	10	13470 (6579/6891)	RR 0.96 (0.91/1.03)	⊕⊕⊕⊝ Moderate 2	It is unclear whether CP reduces mortality compared to controls	
Elbadawi [14]	6	1307 (737/550)	OR 0.83 (0.58/1.18)	⊕⊕⊕⊝ Moderate 2	It is unclear whether CP reduces mortality compared to controls	
Gupta [15]	12	13204 (6715/6489)	RR 0.81 (0.65/1.02)	⊕⊕⊕⊝ Moderate 2	It is unclear whether CP reduces mortality compared to controls	
Janiaud [16]	9	1384 (758/626)	RR 1.02 (0.92/1.12)	⊕⊕⊕⊝ Moderate 6	It is unclear whether CP reduces mortality compared to controls	
Klassen [21]	10	1425 (771/654)	OR 0.76 (0.54/1.09)	From moderate ⊕⊕⊕⊝ to high ⊕⊕⊕⊕ 3	It is unclear whether CP reduces mortality compared to controls	
Meher [24]	2	187 (94/93)	RR 0.60 (0.33/1.10)	⊕⊕⊕⊝ Moderate 6	It is unclear whether CP reduces mortality compared to controls	
Prasad [27]	8	1336 (730/606)	OR 0.85 (0.61/1.18)	⊕⊕⊕⊝ Moderate 2	It is unclear whether CP reduces mortality compared to controls	
Wang M [33]	2	167 (81/86)	OR 0.40 (0.14/1.11)	⊕⊕⊕⊝ Moderate 6	It is unclear whether CP reduces mortality compared to controls	
Yuwono Soeroto [37]	6	1142 ()647/495)	RR 0.85 (0.71/1.02)	⊕⊕⊕⊝ Moderate 2	It is unclear whether CP reduces mortality compared to controls	
	Non-RCTs only					
Bansal [11]	9	4087 (1278/2809)	OR 0.78 (0.65/0.94)	⊕⊕⊝⊝ Low 1	CP reduces mortality compared to controls	
Barreira [12]	6	2033 (328/1705)	RR 0.56 (0.39/0.81)	⊕⊕⊝⊝ Low 1	CP reduces mortality compared to controls	
Klassen [21]	20	11467 (3150/8317)	OR 0.57 (0.45/0.72)	From moderate ⊕⊕⊕⊝ To high ⊕⊕⊕⊕ 3	CP reduces mortality compared to controls	
Meher [24]	4	273 (73/200)	OR 0.48 (0.17/1.36)	⊕⊝⊝⊝ Very low 4	It is unclear whether CP reduces mortality compared to controls	
Prasad [27]	13	8267 (2621/5646)	OR 0.66 (0.53/0.82)	⊕⊕⊝⊝ Low 1	CP reduces mortality compared to controls	
Wang M [33]	11	7779 (1649/6130)	RR 0.59 (0.53/0.66)	⊕⊝⊝⊝ Very low 5	CP reduces mortality compared to controls	
Yuwono Soeroto [37]	12	4515 (1521/2994)	RR 0.48 (0.34/0.70)	⊕⊕⊕⊝ Moderate 2	CP reduces mortality compared to controls	
	Subgroup analysis: High antibody titre					
Aviani [10]	High titre (>640) -3 studies (2 RCTs) in severely ill pts -2 studies (1 RCT) in critical ill pts	1186 (98/1088)388 (43/345)	RR 0.42 (0.22/0.78)RR 0.72 (0.46/1.12)	⊕⊕⊕⊝ Moderate 2 ⊕⊕⊝⊝ Low 7	-In pts with severe illness, high titre CP reduces mortality compared to controls. The reduction in mortality in study with lower antibody titre (neutralizing titre ≤ 1:320) was less marked (RR 0.80, 95% CI 0.47/1.34). -In pts with critical illness, it is unclear whether CP reduces mortality compared to controls	

Barreira [12]	High titre (>1:297) (4 studies, 2 RCTs)	650 (329/321)	RR 0.68 (0.44/1.04)	⊕⊕⊝⊝ Low 7	It is unclear whether high titre CP reduces mortality compared to controls. In studies with lower titre (<1:297) CP the RR was higher: 0.85 (0.58/1.25)	
Cruciani [17]	High titre, 3 RCTs	374 (170/174)	RD −0.06 (−0.12/0.00)	⊕⊕⊕⊝ Moderate 2	High titre CP reduces mortality compared to controls	
De Candia [13]	High titre (different cut-off) (14 studies, 7 RCTs)	20744 (11711/9033)	RR 0.93 (0.88/0.99)	⊕⊕⊕⊝ Moderate 2	High titre CP reduces mortality compared to controls	
Klassen [21]	2 non-RCTs	1125 (534 high titre, 591 lower titre)	22% high titre CP, 29% lower titre	⊕⊕⊝⊝ Low 7	Mortality higher in those receiving lower CP titre transfusion	
	Subgroup analysis: early CP transfusion					
Barreira [12]	3 studies (2 RCTs)	2118 (416/1702)	RR 0.71 (0.53/0.96)	⊕⊕⊕⊝ Moderate 2	Mortality was reduced in pts receiving early (within 7 days) CP transfusion compared to controls. In pts receiving late (> 7 days) CP transfusion mortality was similar to that observed in control group (RR 0.60, 95% CI 0.30/1.17)	
De Candia [13]	11 studies (5 RCTs)	19007 (8018/10989)	OR 0.72 (0.68/0.77)	⊕⊕⊕⊝ Moderate 2	Mortality was reduced in pts receiving early (within 3 days) CP transfusion compared to controls. In pts receiving late (> 7 days) CP transfusion mortality was similar to that observed in control group (OR 0.94, 95% CI 0.86/1.04)	
Klassen [21]	8 studies (3 RCTs)	1561 (656/905)	OR 0.44 (0.32/0.61)	⊕⊕⊕⊝ Moderate 2	Mortality reduction associated with convalescent plasma transfusion was greater in studies that transfused patients within 3 days of hospital admission (OR, 0.44; 95% CI 0.32–0.61) compared with studies that transfused patients more than 3 days after hospital admission (OR, 0.79; 95% CI 0.62/0.98)	
Piechotta [26]	1 RCT (Recovery trial)	4466	RR 0.93 (0.84/1.02)	⊕⊕⊕⊝ Moderate 2	It is unclear whether early CP transfusion (within 7 days of symptoms onset) reduces mortality compared to controls. Transfusion of CP after 7 days of symptoms onset resulted in a RR of 1.04 (0.95/1.15)	
	Subgroup analysis according to severity of infection					
	-Moderate COVID-19					
Barreira [12]	3 (2 RCTs)	545 (275/272)	RR 0.96 (0.62/1.48)	⊕⊕⊝⊝ Low 1	It is unclear whether CP transfusion reduces mortality in pts with moderate illness	
Kim [20]	3 non-RCTs	Not available	OR 0.67 (0.16/2.74)	⊕⊕⊝⊝ Low 8	It is unclear whether CP transfusion reduces mortality in pts with moderate illness	
Piechotta (from moderate to severe) [26]	1 RCT	77 (36/41)	RR 0.98 (0.68/1.41)	⊕⊕⊝⊝ Low 9	It is unclear whether CP transfusion reduces mortality in pts with moderate/severe illness	
Yuwono Soeroto (from moderate to severe) [37]	6 (3 RCTs)	938 (430/508)	RR 0.51 (0.26/1.02)	⊕⊕⊝⊝ Low 8	It is unclear whether CP transfusion reduces mortality in pts with moderate/severe illness	
	-Severe/ critical COVID-19					
Aviani (Severe vs. critical) [10]	4 (1 RCT)	166 (80/86)	RR 4.64 (2.12/10.0)	⊕⊕⊝⊝ Low 7	In pts receiving CP, mortality higher in critical ill pts compared to severely illness	
Barreira [12]	4 (2 RCTs)	1889 (240/1649)	RR 0.84 (0.54/1.32)	⊕⊕⊕⊝ Moderate 2	It is unclear whether CP transfusion reduces mortality in pts with severe/critical illness	
Wardhani [35]	9 trials (3 RCTs)	4164 (1458/2706)	OR 1.32 (1.09/1.60)	⊕⊕⊕⊝ Moderate 2	In pts with severe disease, mortality higher in controls compared to CP recipients	
Wenjing [36]	Severe: 4 trials (1 RCT)Critical: 3 trials (1 RCT)	1420 (168/1252)171 (78/93)	RR 0.54 (0.36/0.80)RR 0.72 (0.35/1.47)	⊕⊕⊕⊝Moderate 2⊕⊝⊝⊝ very low 5	CP reduces mortality in pts with severe illness It is unclear whether CP transfusion reduces mortality in pts with critical illness	

Yuwono Soeroto [37]	13 (6 RCTs)	4899 (1718/3181)	RR 0.68 (0.51/0.91)	⊕⊕⊝⊝ Low 8	In pts with severe/critical illness, CP reduces mortality compared to controls	

Footnotes: 
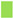
 Favouring CP, 
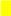
 Not clear effect of CP compared to controls. 1. Downgraded for risk of bias and heterogeneity. 2. Downgraded for risk of bias. 3. This judgment was based on the consistency of the results between RCTs and matched control studies and the corroborating evidence from dose–response studies and other uncontrolled case data. In aggregating data from all controlled studies, the meta-analyses provided precise estimates, did not demonstrate substantial heterogeneity, and demonstrated no evidence of publication bias. The inherent limitations of the included studies rendered the certainty of evidence judgment to be moderate to high. 4. Downgraded for risk of bias, heterogeneity and publication bias. 5. Downgraded for risk of bias, imprecision and heterogeneity. 6. Downgraded for imprecision. 7. Downgraded for risk of bias and imprecision. 8. Downgraded twice for risk of bias (confounding and publication bias). 9. Downgraded for imprecision and clinical heterogeneity.

**Table 4 diagnostics-11-01663-t004:** Effects of convalescent plasma on viral clearance, clinical improvement and length of hospital stay.

Review [Reference]	No. Studies	No. Subjects (CP/Controls)	Effect Size (RR, OR, HR or MD) and 95% CIs	GRADE Assessment	Comment	Effect Direction
Outcome: viral clearance						
-on day 3						
Barreira [12]	4 trials (3 RCTs)	552 (276/276)	RR 0.61 (0.38/0.98)	⊕⊕⊕⊝ Moderate 1	Compared to standard treatment, CP increases rate of viral clearance after 3 days	
Peng [25]	3 trials (2 RCTs)	128 (63/65)	OR 26.21 (4.36/157.66)	⊕⊕⊕⊝ Moderate 2	CP increases viral clearance compared to controls	
Piechotta [26]	4 RCTs	552 (279/273)	RR 1.73 (0.98/3.04)	⊕⊕⊕⊝ Moderate 1	In patients with moderate/severe disease, it is unclear whether CP increases rate of viral clearance compared to controls	
Prasad [27]	2 RCTs	551 (282/269)	OR 3.62 (0.43/30.49)	⊕⊕⊝⊝ Low 3	It is unclear whether CP increases rate of viral clearance compared to controls	
Wang Y [34]	2 trials (1 RCT)	108 (53/55)	RR 2.47 (1.70/3.57)	⊕⊕⊝⊝ Low 4	CP increases viral clearance compared to controls	
Zhang [38]	2 trials (1 RCT)	108 (53/55)	RR 2.55 (1.76/3.70)	⊕⊕⊝⊝ Low 4	CP increases viral clearance compared to controls	
-on day 7–14						
Kim [20]	3 trials (1 RCT)	NA	OR 11.39 (3.91/33.18)	⊕⊕⊝⊝ Low 5	On day 14, CP showed significantly higher viral clearance rate compared to standard supportive therapy	
Piechotta [26]	2 RCTs	149 (79/70)	RR 1.59 (0.74/3.43)	⊕⊕⊝⊝ Low 3	On day 14, it is unclear whether CP increases viral clearance rate compared to standard supportive therapy	
Outcome: clinical improvement						
Elbadawi [14]	3 RCTs	267 (130/137)	OR 1.31 (0.78/2.22)	⊕⊕⊝⊝ Low 5	It is unclear whether CP increases rate of clinical improvement compared to controls	
Gupta [15]	6 RCTs	1106 (616/490)	RR 1.02 (0.82/1.28)	⊕⊕⊕⊝ Moderate 6	It is unclear whether CP increases rate of clinical improvement compared to controls after 7, 14 and 28 days from administration	
Peng [25]	4 trials (2 RCTs)	404 (144/260)	OR 1.54 (0.79/3.01)	⊕⊕⊝⊝ Low 5	It is unclear whether CP increase rate of clinical improvement compared to controls	
Piechotta [26]	1 RCT	77 (36/41)	RR 1.10 (0.83/1.48)	⊕⊕⊕⊝ Moderate 2	In patients with moderate/severe disease, it is unclear whether CP increases rate of clinical improvement (liberation from supplemental oxygen) compared to controls	
Prasad [27]	3 RCTs	421 (322/199)	OR 1.07 (0.86/1.34)	⊕⊕⊝⊝ Low 5	It is unclear whether CP increases rate of clinical improvement compared to controls	
Sarkar [29]	7 trials (2 RCTs)	5454	OR 0.44 (0.25/0.77)	⊕⊕⊕⊝ Moderate 6	CP increase rate of clinical improvement compared to controls	
Talaie [31]	3 trials (2 RCTs)	Not available	RR 1.41 (1.01/1.98)	⊕⊕⊝⊝ Low 5	CP increase rate of clinical improvement compared to controls	
Vegivinti [32]	7 trials (2 RCTs)	1581	OR 2.02 (1.54/2.65)	⊕⊕⊝⊝ Low 5	CP increase rate of clinical improvement compared to controls	
Wang M [33]	2 RCTs	189 (95/94)	OR 1.21 (0.68/2.16)	⊕⊕⊕⊝ Moderate 2	It is unclear whether CP increase rate of clinical improvement compared to controls	
Outcome: length of hospital stay						
Barreira [12]	3 trials (1 RCT)	2221 (488/1733)	MD 1.94 (−3.69/7.58)	⊕⊕⊝⊝ Low 5	It is unclear whether CP decreases length of hospital stay compared to controls	
Janiaud [16]	4 RCTs (2 published as preprints)	603 (361/242)	HR 1.07 (0.79/1.45)	⊕⊕⊝⊝ Low 4	It is unclear whether CP decreases length of hospital stay compared to controls	
Peng [25]	6 trials (1 RCT)	2101 (366/1735)	MD 0.84 (−3.35/5.02)	⊕⊕⊝⊝ Low 5	It is unclear whether CP decreases length of hospital stay compared to controls	
Piechotta [26]	5 RCTs	683 (401/282)	HR 1.15 (0.95/1.40)	⊕⊕⊕⊝ Moderate 2	It is unclear whether CP decreases length of hospital stay compared to controls	
Prasad [27]	4 trials (3 RCTs)	2602 (1365/1237)	MD 0.12 (−1.69/1.93)	⊕⊕⊕⊝ Moderate 7	It is unclear whether CP decreases length of hospital stay compared to controls	
Vegivinti [32]	6 trials (2 RCTs)	2157	MD −0.5 (−3.1/2.1)	⊕⊕⊝⊝ Low 5	It is unclear whether CP decreases length of hospital stay compared to controls	

Footnotes: 
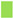
 Favouring CP, 
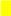
 Not clear effect of CP compared to controls. 1. Downgraded for heterogeneity. 2. Downgraded for imprecision. 3. Downgraded for inconsistency and imprecision. 4. Downgraded for ROB and imprecision. 5. Downgraded for ROB and inconsistency. 6. Downgraded for ROB. 7. Downgraded for inconsistency.

**Table 5 diagnostics-11-01663-t005:** Adverse reactions related to convalescent plasma transfusion.

Review [Reference]	No. Studies	No. Events/No. Patients in CP and Controls		Effect size (RR or RD) and 95% CIs	GRADE Assessment	Comment	Effect Direction
Outcome: Adverse Events (AE)							
-Serious AE							
Gupta [15]	11 RCTs	200/6164 (3.2%)	161/5826 (2.87)	RR 1.14 (0.93/1.40)	⊕⊕⊕⊝ Moderate 1	Serious AE were rare and with similar frequency in CP and controls	
Janiaud [16]	3 RCTs	60/329 (18.2%)	26/191 (13.6%)	RR 0.97 (0.36/2.63)	⊕⊕⊝⊝ Low 2	Serious AE had similar frequency in CP and controls	
Juul [18]	3 RCTs	46/249 (18.4%)	49/262 (18.7%)	RR 0.93 (0.36/2.63)	⊕⊕⊕⊝ Moderate 1	Serious AE had similar frequency in CP and controls	
Franchini [22]	9 RCTs	106/853 (12.4%)	52/616 (8.4%)	RD 0.00 (−0.03/0.03)	⊕⊕⊕⊝ Moderate 1	Serious AE had similar frequency in CP and controls	
Piechotta [26]	2 RCTs	60/206 (29.1%)	26/148 (17.5%)	RR 1.73 (0.98/3.04)	⊕⊕⊕⊝ Moderate 3	Serious AE were more common in CP compared to controls (29.1 vs 17.5%) but the difference was not statistically significant	
-Overall AE							
Franchini [22]	8 RCTs	1692/5848 (28.9%)	1535/5471 (28.0%)	RD 0.01 (−0.02/0.03)	⊕⊕⊕⊝ Moderate 1	Overall AE had similar frequency in CP and controls (28%)	
Piechotta [26]	1 RCT	153/228 (67%)	66/104 (63.4%)	RR 1.06 (0.89/1.26)	⊕⊕⊕⊝ Moderate 3	Overall AE had similar frequency in CP and controls (>60%). Likewise, grade 3–4 AE had similar frequency in CP recipients and controls (8.5 and 6.3%, respectively)	

Footnotes: 
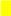
 Not clear effect of CP compared to controls. 1. Downgraded for ROB. 2. Downgraded for ROB and imprecision. 3. Downgraded for imprecision.

## Data Availability

Not applicable.

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
