# Peer review of "Safety and Efficacy of Convalescent Plasma in COVID-19: An Overview of Systematic Reviews"

_diagnostics, 2021, doi:10.3390/diagnostics11091663_

Round 1

Reviewer 1 Report

The present study have remarkable points but some aspects should be reviewed.

Strong points: No other umbrella reviews have been published to date and the heterogeneity of data supports such a review.

Appropriate tittle (with PICO elements) and methodology for umbrella review were well applied: data extraction, use of AMSTAR-2 for assessment, number of reviewers, discussion in discrepancies, …

Paper was nicely written and clearly structured.

Points that should be reviewed:

  • Results should include Forest plot to better visualize OR or RR results.
  • Main results Column in Table 1 should be synthesized for better visualization of data.

  • Table 1: In intervention change standard of care for Standard care

  • There are redundancy of data in table 1 and 3 (RR/OR and 95% CI). Sometimes data are not coincident so, they must be checked and corrected (Example: Aviani)
  • Table 3: Inclusion of a head column Type of evidence (ES) to indicate RR or OR mean should be considered. It should include a column with p values. Information in Comments should be synthetized with an arrow in mortality to indicate a reduction in mortality. It is not necessary to specify “CP reduce … “ because the intervention is always CP, nor “compared to controls”. Other longer comment should also be synthesized when subgroups analysis are compared in base of lower antibody titers o late transfusion. The table should only contain a summary of data.
  • The previous observations should also be applied to tables 4 and 5.
  • Reviews 17 and 22 were not included in Table 4 and that was not justified.

Author Response

Dear Editor,

Many thanks for your positive comments on our manuscript. We also thank the Reviewers for the detailed suggestions on how to improve the manuscript All their suggestions were carefully considered in this revision and all changes in the manuscript are clearly showed.

Sincerely yours,

Massimo Franchini

Reviewer 1

The present study have remarkable points but some aspects should be reviewed.

Strong points: No other umbrella reviews have been published to date and the heterogeneity of data supports such a review.

Appropriate tittle (with PICO elements) and methodology for umbrella review were well applied: data extraction, use of AMSTAR-2 for assessment, number of reviewers, discussion in discrepancies, …

Paper was nicely written and clearly structured.

Points that should be reviewed:

  • Results should include Forest plot to better visualize OR or RR results.

Answer: The aim of an Umbrella Review is to provide a summary of existing research syntheses related to a given topic or question and not to re-synthesize, for example, the results of existing reviews or syntheses with meta-analysis  and  related forest-plots. This will be methodologically  incorrect due to the number of overlapping trials included in each meta-analysis. On the other hand it is problematic to presents the quantitative results of each systematic review with forest plots: in our umbrella review, 89 comparisons were summarized in tabular form, and it will be technically challenging  presenting these data using 89 forest plots.

  • Main results Column in Table 1 should be synthesized for better visualization of data.

Answer: done.

  • Table 1: In intervention change standard of care for Standard care

Answer: done.

  • There are redundancy of data in table 1 and 3 (RR/OR and 95% CI). Sometimes data are not coincident so, they must be checked and corrected (Example: Aviani)

Answer: done.

  • Table 3: Inclusion of a head column Type of evidence (ES) to indicate RR or OR mean should be considered. It should include a column with p values. Information in Comments should be synthetized with an arrow in mortality to indicate a reduction in mortality. It is not necessary to specify “CP reduce … “ because the intervention is always CP, nor “compared to controls”. Other longer comment should also be synthesized when subgroups analysis are compared in base of lower antibody titers o late transfusion. The table should only contain a summary of data.

Answers: We have now specified in the head of the columns that effect size are RR, OR, and RD are presented, as specified in each cell.

We believe that Confidence intervals are preferable to p-values, as they tell us the range of possible effect sizes compatible with the data. p-values simply provide a cut-off beyond which we assert that the findings are 'statistically significant' 

We believe that a brief statements, as those reported in the SOT tables, may help the interpretation of data by readers more than a simple arrow.

  • The previous observations should also be applied to tables 4 and 5.

Answers: done

  • Reviews 17 and 22 were not included in Table 4 and that was not justified.

Answer: table 4 reports virological outcomes, which are not included in ref 17 and 22. For this reason, they have been excluded.

Reviewer 2 Report

This is an interesting umbrella review, written by researchers well-versed in the topic and in the methodology.

I have just some concerns.

1) In the abstract, line 25, lack of funding sources is in 26/29 studies and not in 26/23.

2) Line 47, introduction: a reference would be useful for the general use of CP from inception to the present.

3) Line 49: it is debatable that the CP is the most used and studied antibody-based therapy, considering the increasing role on monoclonal antibodies.

4) Methodology. How was subgroup analysis performed? Did you simply abstract data from SRs regarding the specific subset of patients.

5) Please provide a more specific definition of high-titre CP.

6) Line 179, the passage is a little bit convoluted, based is repeated twice.

7) Line 191, Cochrane with the first C in capital letter.

8) Table 3, the CI is reported inconsistently (hyphen or slash: please uniform).

Author Response

Dear Editor,

Many thanks for your positive comments on our manuscript. We also thank the Reviewers for the detailed suggestions on how to improve the manuscript All their suggestions were carefully considered in this revision and all changes in the manuscript are clearly showed.

Sincerely yours,

Massimo Franchini

Reviewer 2

This is an interesting umbrella review, written by researchers well-versed in the topic and in the methodology.

I have just some concerns.

1) In the abstract, line 25, lack of funding sources is in 26/29 studies and not in 26/23.

Answer: done.

2) Line 47, introduction: a reference would be useful for the general use of CP from inception to the present.

Answer: Done.

3) Line 49: it is debatable that the CP is the most used and studied antibody-based therapy, considering the increasing role on monoclonal antibodies.

Answer: I have mitigated this sentence.

4) Methodology. How was subgroup analysis performed? Did you simply abstract data from SRs regarding the specific subset of patients.

Answer: Yes. Subgroup analysis data were extracted by SRs.

5) Please provide a more specific definition of high-titre CP.

Answer: it is not possible to give a unique definition of high-titre CP as it greatly differs from study to study, depending on the type of test used (serologic or PRNT or equivalent).

6) Line 179, the passage is a little bit convoluted, based is repeated twice.

Answer: done.

7) Line 191, Cochrane with the first C in capital letter.

Answer: done.

8) Table 3, the CI is reported inconsistently (hyphen or slash: please uniform).

Answer: done.

Round 2

Reviewer 2 Report

The authors have replies sufficiently to the issues raised The paper may be published in the present form.